# SesameBERT: Attention for Anywhere

## Abstract

Fine-tuning with pre-trained models has achieved exceptional results for many language tasks. In this study, we focused on one such self-attention network model, namely BERT, which has performed well in terms of stacking layers across diverse language-understanding benchmarks. However, in many downstream tasks, information between layers is ignored by BERT for fine-tuning. In addition, although self-attention networks are well-known for their ability to capture global dependencies, room for improvement remains in terms of emphasizing the importance of local contexts. In light of these advantages and disadvantages, this paper proposes SesameBERT, a generalized fine-tuning method that (1) enables the extraction of global information among all layers through Squeeze and Excitation and (2) enriches local information by capturing neighboring contexts via Gaussian blurring. Furthermore, we demonstrated the effectiveness of our approach in the HANS dataset, which is used to determine whether models have adopted shallow heuristics instead of learning underlying generalizations. The experiments revealed that SesameBERT outperformed BERT with respect to GLUE benchmark and the HANS evaluation set.

## 1 Introduction

In recent years, unsupervised pretrained models have dominated the field of natural language processing (NLP). The construction of a framework for such a model involves two steps: pretraining and fine-tuning. During pretraining, an encoder neural network model is trained using large-scale unlabeled data to learn word embeddings; parameters are then fine-tuned with labeled data related to downstream tasks.

Traditionally, word embeddings are vector representations learned from large quantities of unstructured textual data such as those from Wikipedia corpora (Mikolov et al., 2013). Each word is represented by an independent vector, even though many words are morphologically similar. To solve this problem, techniques for contextualized word representation (Peters et al., 2018; Devlin et al., 2019) have been developed; some have proven to be more effective than conventional word-embedding techniques, which extract only local semantic information of individual words. By contrast, pretrained contextual representations learn sentence-level information from sentence encoders and can generate multiple word embeddings for a word. Pretraining methods related to contextualized word representation, such as BERT (Devlin et al., 2019), OpenAI GPT (Radford et al., 2018), and ELMo (Peters et al., 2018), have attracted considerable attention in the field of NLP and have achieved high accuracy in GLUE tasks such as single-sentence, similarity and paraphrasing, and inference tasks (Wang et al., 2019). Among the aforementioned pretraining methods, BERT, a state-of-the-art network, is the leading method that applies the architecture of the Transformer encoder, which outperforms other models with respect to the GLUE benchmark. BERT's performance suggests that self-attention is highly effective in extracting the latent meanings of sentence embeddings.

This study aimed to improve contextualized word embeddings, which constitute the output of encoder layers to be fed into a classifier. We used the original method of the pretraining stage in the BERT model. During the fine-tuning process, we introduced a new architecture known as Squeeze and Excitation alongside Gaussian blurring with symmetrically SAME padding ("SESAME" hereafter). First, although the developer of the BERT model initially presented several options for its use, whether the selective layer approaches involved information contained in all layers was unclear. In a previous study, by investigating relationships between layers, we observed that the Squeeze and Excitation method (Hu et al., 2018) is key for focusing on information between layer weights. This

method enables the network to perform feature recalibration and improves the quality of representations by selectively emphasizing informative features and suppressing redundant ones. Second, the self-attention mechanism enables a word to analyze other words in an input sequence; this process can lead to more accurate encoding. The main benefit of the self-attention mechanism method is its high ability to capture global dependencies. Therefore, this paper proposes the strategy, namely Gaussian blurring, to focus on local contexts. We created a Gaussian matrix and performed convolution alongside a fixed window size for sentence embedding. Convolution helps a word to focus on not only its own importance but also its relationships with neighboring words. Through such focus, each word in a sentence can simultaneously maintain global and local dependencies.

We conducted experiments with our proposed method to determine whether the trained model could outperform the BERT model. We observed that SesameBERT yielded marked improvement across most GLUE tasks. In addition, we adopted a new evaluation set called HANS (McCoy et al., 2019), which was designed to diagnose the use of fallible structural heuristics, namely the lexical overlap heuristic, subsequent heuristic, and constituent heuristic. Models that apply these heuristics are guaranteed to fail in the HANS dataset. For example, although BERT scores highly in the given test set, it performs poorly in the HANS dataset; BERT may label an example correctly not based on reasoning regarding the meanings of sentences but rather by assuming that the premise entails any hypothesis whose words all appear in the premise (Dasgupta et al., 2018). By contrast, SesameBERT performs well in the HANS dataset; this implies that this model does not merely rely on heuristics. In summary, our final model proved to be competitive on multiple downstream tasks.

## 2 RELATED WORK

### 2.1 UNSUPERVISED PRETRAINING IN NLP

Most related studies have used pretrained word vectors (Mikolov et al., 2013; Pennington et al., 2014) as the primary components of NLP architectures. This is problematic because word vectors capture semantics only from a word's surrounding text. Therefore, a vector has the same embedding for the same word in different contexts, even though the word's meaning may be different.

Pretrained contextualized word representations overcome the shortcomings of word vectors by capturing the meanings of words with respect to context. ELMo (Peters et al., 2018) can extract context-sensitive representations from a language model by using hidden states in stacked LSTMs. Generative pretraining (Radford et al., 2018) uses the "Transformer encoder" rather than LSTMs to acquire textual representations for NLP downstream tasks; however, one limitation of this model is that it is trained to predict future left-to-right contexts of a unidirectional nature. BERT (Devlin et al., 2019) involves a masked language modeling task and achieves high performance on multiple natural language-understanding tasks. In BERT architecture, however, because the output data of different layers encode a wide variety of information, the most appropriate pooling strategy depends on the case. Therefore, layer selection is a challenge in learning how to apply the aforementioned models.

### 2.2 SQUEEZE AND EXCITATION

The Squeeze and Excitation method was introduced by Hu et al. (2018), who aimed to enhance the quality of representations produced by a network. Convolutional neural networks traditionally use convolutional filters to extract informative features from images. Such extraction is achieved by fusing the spatial and channel-wise information of the image in question. However, the channels of such networks' convolutional features have no interdependencies with one another. The network weighs each of its channels equally during the creation of output feature maps. Through Squeeze and Excitation, a network can take advantage of feature recalibration and use global information to emphasize informative features and suppress less important ones.

### 2.3 LOCALNESS MODELING

The self-attention network relies on an attention mechanism to capture global dependencies without considering their distances by calculating all the positions in an input sequence. Our Gaussian-blurring method focuses on learning local contexts while maintaining a high ability to capture long-range dependencies. Localness modeling was considered a learnable form of Gaussian bias (Yang

et al., 2019) in which a central position and dynamic window are predicted alongside intermediate representations in a neural network. However, instead of using Gaussian bias to mask the logit similarity of a word, we performed Gaussian bias in the layer after the embedding layer to demonstrate that performing element-wise operations in this layer can improve the model performance.

## 2.4 DIAGNOSING SYNTACTIC HEURISTICS

A recent study (McCoy et al., 2019) investigated whether neural network architectures are prone to adopting shallow heuristics to achieve success in training examples rather than learning the underlying generalizations that need to be captured. For example, in computer vision, neural networks trained to recognize objects are misled by contextual heuristics in cases of monkey recognition (Wang et al., 2017). For example, in the field of natural language inference (NLI), a model may predict a label that contradicts the input because the word "not", which often appears in examples of contradiction in standard NLI training sets, is present (Naik et al., 2018; Carmona et al., 2018). In the present study, we aimed to make SesameBERT robust with respect to all training sets. Consequently, our experiments used HANS datasets to diagnose some fallible structural heuristics presented in this paper (McCoy et al., 2019).

## 3 METHODS

We focused on BERT, which is the encoder architecture of a multilayer Transformer (Vaswani et al., 2017), featuring some improvements. The encoder consists of L encoder layers, each containing two sublayers, namely a multihead self-attention layer and a feed-forward network. The multihead mechanism runs through a scaled dot product attention function, which can be formulated by querying a dictionary entry with key value pairs (Miller et al., 2016). The self-attention input consists of a query $\boldsymbol{Q} \in \mathbb{R}^{l \times d}$, a key $\boldsymbol{K} \in \mathbb{R}^{l \times d}$, and a value $\boldsymbol{V} \in \mathbb{R}^{l \times d}$, where $l$ is the length of the input sentence, and $d$ is the dimension of embedding for query, key and value. For subsequent layers, $\boldsymbol{Q}$, $\boldsymbol{K}$, $\boldsymbol{V}$ comes from the output of the previous layer. The scaled dot product attention (Vaswani et al., 2017) is defined as follows:

$$Attention(\boldsymbol{Q}, \boldsymbol{K}, \boldsymbol{V}) = softmax(\frac{\boldsymbol{Q}\boldsymbol{K}^T}{\sqrt{d}}) \cdot \boldsymbol{V} \qquad (1)$$

The output represents the multiplication of the attention weights $\boldsymbol{A}$ and the vector $\boldsymbol{v}$, where $\boldsymbol{A} = softmax(\frac{\boldsymbol{Q}\boldsymbol{K}^T}{\sqrt{d}}) \in \mathbb{R}^{l \times l}$. The attention weights $A_{i,j}$ enabled us to better understand about the importance of the $i$-th key-value pairs with respect to the $j$-th query in generating the output (Bahdanau et al., 2015). During fine-tuning, We used the output encoder layer from the pretrained BERT model to create contextualized word embeddings and feed these embeddings into the model. Although several methods have been developed for extracting contextualized embeddings from various layers, we believed that these methods had substantial room for improvement. Therefore, we used Squeeze and Excitation to solve the aforementioned problem.

## 3.1 SQUEEZE AND EXCITATION

In this study, we proposed the application of Squeeze and Excitation (Hu et al., 2018); its application to the output of the encoder layer was straightforward once we realized that the number of channels was equivalent to the number of layers. Therefore, we intended to use the term channels and layers interchangeably.

First, we defined $\mathbf{U}_{:,:,k}$ as the output of the $k$-th encoder layer, for all $1 \leq k \leq n$. We wanted to acquire global information from between the layers before feeding the input into the classifier; therefore, we concatenated all the output from each encoder layer to form the feature maps $\mathbf{U} \in \mathbb{R}^{l \times d \times n}$. In the squeeze step, by using global average pooling on the $k$th layer, we were able to squeeze the global spatial information into a layer descriptor. In other words, we set the $k$th layer's output of the squeeze function as $\mathbf{Z}_{:,:,k}$.

$$\mathbf{Z}_{:,:,k} = f_{sq}(\mathbf{U}_k) = \frac{1}{l \times d} \sum_{i=1}^{l} \sum_{j=1}^{d} U_{i,j,k} \qquad (2)$$

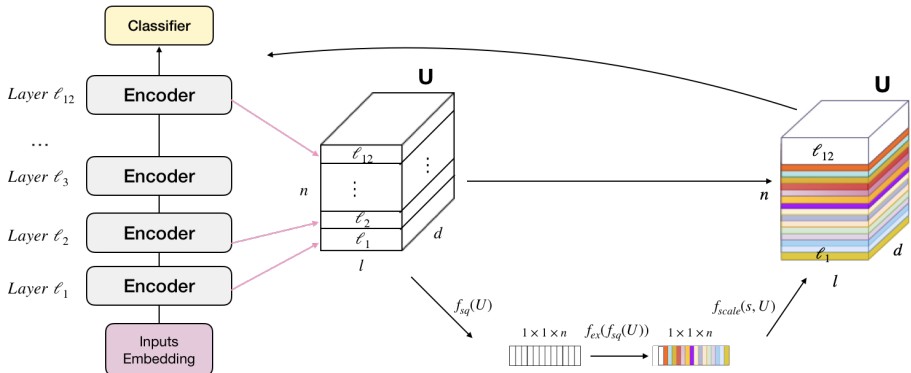

Figure 1: We extracted the output from each layer of the encoders and concatenated all the layers to form a three-dimensional tensor $\mathbf{U}$. We then performed Squeeze $f_{sq}(\mathbf{U})$ and Excitation $f_{ex}(f_{sq}(\mathbf{U}))$ to obtain the weight of each output layer. Finally, we fed the weighted average of all layers into the classifier. In this work we employed $n = 12$ attention layers.

In the excitation step, we aimed to fully capture layer-wise dependencies. This method uses the layer-wise output of the squeeze operation $f_{sq}$ to modulate interdependencies of all layers. Excitation is a gating mechanism with a sigmoid activation function that contains two fully connected layers. Let $\boldsymbol{W}_1$ and $\boldsymbol{W}_2$ be the weights of the first and second fully connected layers, respectively, and let $r$ be the bottleneck in the layer excitation that encodes the layer-wise dependencies; therefore, $\boldsymbol{W}_1 \in \mathbb{R}^{n \times \frac{n}{r}}$, and $\boldsymbol{W}_2 \in \mathbb{R}^{\frac{n}{r} \times n}$. The excitation function $f_{ex}$:

$$s = f_{ex}(z) = \sigma(ReLU(z, \boldsymbol{W}_1), \boldsymbol{W}_2) \tag{3}$$

where $z$ is the vector squeezed from tensor $\mathbf{Z}$.

Finally, we rescaled the output $\mathbf{Z}_{:,:,k}$ by multiplying it by $s_k$. The rescaled output is deonted as $\widetilde{\boldsymbol{u}}_k$. The scaling function $f_{scale}$ is defined as follows:

$$\widetilde{\boldsymbol{u}}_k = f_{scale}(\boldsymbol{s}_k, \mathbf{U}_{:,:,k}) \tag{4}$$

We concatenated all rescaled outputs from all encoder layers to form our rescaled feature maps $\widetilde{\boldsymbol{u}}$. The architecture is shown in Figure 1. We then extracted layers from the rescaled feature maps, or calculated a weighted average layer $\widetilde{\boldsymbol{u}}_{avg}$.

$$\widetilde{\boldsymbol{u}}_{avg} = \frac{\sum_{k=1}^{n} f_{scale}(\boldsymbol{s}_k, \mathbf{U}_{:,:,k})}{\sum_{k=1}^{n} \boldsymbol{s}_k} \tag{5}$$

## 3.2 GAUSSIAN BLURRING

Given an input sequence $X = \{x_1, x_2, ..., x_l\} \in \mathbb{R}^{l \times d}$, the model transformed it into queries $\boldsymbol{Q}$, keys $\boldsymbol{K}$, and values $\boldsymbol{V}$, where $\boldsymbol{Q}, \boldsymbol{K}$, and $\boldsymbol{V} \in \mathbb{R}^{l \times d}$. Multihead attention enabled the model to jointly attend to information from different representation subspaces at different positions. Thus, the three types of representations are split into h subspaces of size $\frac{d}{h}$ to attend to different information. For example, $\boldsymbol{Q} = (\boldsymbol{Q}^1, \boldsymbol{Q}^2, ..., \boldsymbol{Q}^h)$ with $\boldsymbol{Q}^i \in \mathbb{R}^{l \times \frac{d}{h}}$ for all $1 \leq i \leq h$. In each subspace h, the element $o_i^h$ in the output sequence $\boldsymbol{O}^h = (o_1^h, o_2^h, ..., o_l^h)$ is computed as follows:

$$o_i^h = Attention(q_i^h, \boldsymbol{K}^h)\boldsymbol{V}^h \tag{6}$$

where $o_i^h \in \mathbb{R}^{\frac{d}{h}}$.

To capture the local dependency related to each word, we first used a predefined fixed window size $k$ to create a Gaussian blur $g$, where $g \in \mathbb{R}^k$:

$$g(\mathrm{x}; \sigma, k) = exp(\frac{-(\mathrm{x} - \lfloor \frac{k}{2} \rfloor)^2}{2\sigma^2}) \tag{7}$$

where $\sigma$ refers to the standard deviation. Several Gaussian-blurring strategies are feasible for applying convolutional operations to attention outputs.

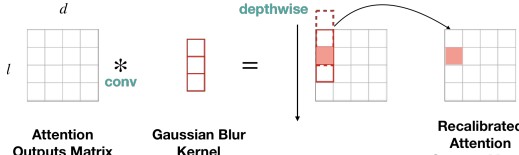

Figure 2: Diagram of a one-dimensional Gaussian blur kernel, which was convoluted through the input dimension. This approach enabled the central word to acquire information concerning neighboring words with weights proportional to the Gaussian distribution.

### 3.2.1 GAUSSIAN BLURRING ON ATTENTION OUTPUTS

The first strategy focuses on each attention output $\boldsymbol{O}^h$. We restrict $\hat{O}^h_{i,j,:}$ to a local scope with a fixed size $k$ centered at the position $i$ and dimension $j$, where $1 \leq j \leq d$, and k can be any odd number between 1 and $l$, expressed as follows:

$$\hat{O}^h_{i,j,:} = [O^h_{i-\lfloor \frac{k}{2} \rfloor,j}, ..., O^h_{i,j}..., O^h_{i+\lfloor \frac{k}{2} \rfloor,j}] \tag{8}$$

We then enhance the localness of $\hat{O}^h_{i,j,:}$ through a parameter-free $1D$ convolution operation with $g$.

$$\widetilde{O}^h_{i,j} = \hat{\boldsymbol{O}}^h_{i,j,:} \cdot g \tag{9}$$

The final attention output is $\widetilde{\boldsymbol{O}}^h$, which is the dot product between the Gaussian kernel and the corresponding input array elements at every position of $\hat{O}^h_{i,j,:}$,

$$\widetilde{\boldsymbol{O}}^h = \boldsymbol{O}^h * g \tag{10}$$

where $*$ is defined as a convolution operation, as illustrated in Figure 2.

More specifically, $\widetilde{O}^h_{i,j}$, the entry of $\widetilde{O}^h$ in the $i$-th row and $j$-th column, equals $blur(O^h_{i,j})$:

$$
\begin{aligned}
\widetilde{O}^h_{ij} &= blur(O^h_{i,j}) \\
&= \sum_{\mathrm{x} \in [-k,k]} g(\mathrm{x}; \sigma, k) O_{i+\mathrm{x},j} \\
&= \sum_{\mathrm{x} \in [-k,k]} g(\mathrm{x}; \sigma, k) \sum_{l} A_{i+\mathrm{x},l} V_{l,j}
\end{aligned}
\tag{11}
$$

### 3.2.2 GAUSSIAN BLURRING ON VALUES

Another option focuses on values V. We applied the aforementioned method again but restrict $\boldsymbol{V}^h$ to a local scope. The final attention output $\widetilde{O}^h$ is denoted as follows:

$$\widetilde{\boldsymbol{O}}^h = Attention(\boldsymbol{Q}^h, \boldsymbol{K}^h)(\boldsymbol{V}^h * g) \tag{12}$$

The difference between the present method and the method of performing Gaussian blurring on attention outputs and values is that the method of performing Gaussian blurring on attention outputs and values places greater emphasis on the interaction of cross-query vectors, whereas the present method focuses on cross-value vectors. Finally, the outputs of the h attention heads are concatenated to form the final output representation $\widetilde{\boldsymbol{O}}$:

$$\widetilde{\boldsymbol{O}} = (\widetilde{\boldsymbol{O}}^1, \widetilde{\boldsymbol{O}}^2, ..., \widetilde{\boldsymbol{O}}^h) \tag{13}$$

where $\widetilde{\boldsymbol{O}} \in \mathbb{R}^{l \times d}$. The multihead mechanism enables each head to capture distinct linguistic input properties (Li et al., 2019). Furthermore, because our model is based on BERT, which builds an encoder framework with a stack of 12 layers, we were able to apply locality modeling to all layers through Squeeze and Excitation. Therefore, we expected that the global information and local properties captured by all layers could be exploited.

## 4 EXPERIMENTS

We evaluated the proposed SesameBERT model by conducting multiple classification tasks. For comparison with the results of a previous study on BERT (Devlin et al., 2019), we reimplemented the BERT model in TensorFlow in our experiments. [1] In addition, we set most of the parameters to be identical to those in the original BERT model, namely, batch size: 16, 32, learning rate: 5e-5, 3e-5, 2e-5, and number of epochs: 3, 4. All of the results in this paper can be replicated in no more than 12 hours by a graphics processing unit with nine GLUE datasets. We trained all of the models in the same computation environment with an NVIDIA Tesla V100 graphics processing unit.

### 4.1 GLUE DATASETS

GLUE benchmark is a collection of nine natural language-understanding tasks, including question-answering, sentiment analysis, identification of textual similarities, and recognition of textual entailment (Wang et al., 2019). GLUE datasets were employed because they are sets of tools used to evaluate the performance of models for a diverse set of existing NLU tasks. The datasets and metrics used for the experiments in this study are detailed in the appendix A.

### 4.2 HANS DATASET

We used a new evaluation set, namely the HANS dataset, to diagnose fallible structural heuristics presented in a previous study (McCoy et al., 2019) based on syntactic properties. More specifically, models might apply accurate labels not based on reasoning regarding the meanings of words but rather by assuming that the $premise$ entails any $hypothesis$ whose words all appear in the premise (Dasgupta et al., 2018; Naik et al., 2018). Furthermore, an instance that contradicts the lexical overlap heuristics in MNLI is likely too rare to prevent a model from learning heuristics. Models may learn to assume that a label is contradictory whenever a negation word is contained in the premise but not the hypothesis (McCoy & Linzen, 2019). Therefore, whether a model scored well on a given test set because it relied on heuristics can be observed. For example, BERT performed well on MNLI tasks but poorly on the HANS dataset; this finding suggested that the BERT model employs the aforementioned heuristics.

The main difference between the MNLI and HANS datasets is their numbers of labels. The MNLI dataset has three labels, namely Entailment, Neutral, and Contradiction. In the HANS dataset, instances labeled as Contradiction or Neutral are translated into non-entailment. Therefore, this dataset has only two labels: Entailment and Non-entailment. The HANS dataset targets three heuristics, namely Lexical overlap, Subsequence, and Constituent, with more details in appendix B. This dataset not only serves as a tool for measuring progress in this field but also enables the visualization of interpretable shortcomings in models trained using MNLI.

### 4.3 RESULTS

#### 4.3.1 GLUE DATASETS RESULTS

This subsection provides the experiment results of the baseline model and the models trained using our proposed method. We performed Gaussian blurring on attention outputs in the experiment. In addition, we employed a batch size of 32, learning rates of 3e-5, and 3 epochs over the data for all GLUE tasks. We fine-tuned the SesameBERT model through 9 downstream tasks in the datasets. For each task, we performed fine-tuning alongside Gaussian blur kernel sigmas 1e-2, 1e-1, 3e-1, and 5e-1 and selected that with the most favorable performance in the dev set. Because GLUE datasets do not distribute labels for test sets, we uploaded our predictions to the GLUE server for evaluation. The results are presented in Table 1; GLUE benchmark is provided for reference. In most tasks, our proposed method outperformed the original BERT-Base model (Devlin et al., 2019). For example, in the RTE and AX datasets, SesameBERT yielded improvements of 1.2% and 1.6%, respectively. We conducted experiments on GLUE datasets to test the effects of Gaussian blurring alongside BERT on the value layer and context layer. Table 2 shows the degrees of accuracy in the dev set. The performance of Gaussian blurring with respect to self-attention layers varied among cases.

---

[1]Our code will be released upon acceptance.

Table 1: Test results in relation to the GLUE benchmark. The metrics for these tasks, shown in appendix A, were calculated using a GLUE score. We compared our SesameBERT model with the original BERT-Base model, ELMo (Peters et al., 2018) and OpenAI GPT (Radford et al., 2018). All results were obtained from `https://gluebenchmark.com/leaderboard`.

|  | BiLSTM+ELMo+Attn | OpenAI GPT | BERT-Base | SesameBERT |
|---|---|---|---|---|
| CoLA | 33.6 | 45.4 | 52.1 | **52.7** |
| SST-2 | 90.4 | 91.3 | 93.5 | **94.2** |
| MRPC | 84.4 | 82.3 | **88.9** | **88.9** |
| STS-B | 72.3 | 80.0 | **85.8** | 85.5 |
| QQP | 63.1 | 70.3 | **71.2** | 70.8 |
| MNLI-m | 74.1 | 82.1 | **84.6** | 83.7 |
| MNLI-mm | 74.5 | 81.4 | 83.4 | **83.6** |
| QNLI | 79.8 | 88.1 | 90.5 | **91.0** |
| RTE | 58.9 | 56.0 | 66.4 | **67.6** |
| AX | 21.7 | - | 34.2 | **35.8** |
| GLUE score | 70.0 | 76.9 | 78.3 | **78.6** |

Table 2: Performance of Gaussian blurring alongside the BERT model. The results were tested on four GLUE datasets, with accuracy as the metric.

|  | MRPC | RTE | QNLI | SST-2 |
|---|---|---|---|---|
| BERT | 86.7 | 65.3 | 88.4 | **92.7** |
| Blur on Value layer | **86.8** | 69.7 | **90.9** | 91.3 |
| Blur on Context layer | 86.5 | **70.4** | 90.8 | 92.0 |

Gong et al. (2019) demonstrated that different layers vary in terms of their abilities to distinguish and capture neighboring positions and global dependency between words. We evaluated the weights learned from all layers. These weights indicated that a heavier weight represents greater importance. The results are shown in appendix C. Because the lower layer represents word embeddings that are deficient in terms of context (Baosong Yang, 2018), the self-attention model in the lower layer may need to encode representations with global context and may struggle to learn localness. Table 3 shows the degree of accuracy predicted by each extracted attention output layer method. The results indicated that the lower layers had lower accuracy.

We performed three ablation studies. First, we examined the performance of our method without blurring; we observed that Squeeze and Excitation helped the higher layer. This trend suggested that higher layers benefit more than do lower layers from Squeeze and Excitation. Second, we analyzed the effect of Gaussian blurring on the context layer. The results revealed that the method with blurring achieved higher accuracy in lower layers. We assumed that capturing short-range dependencies among neighboring words in lower layers is an effective strategy. Even if self-attention models capture long-range dependencies beyond phrase boundaries in higher layers, modeling localness remains a helpful metric. Finally, we observed the direct effects of SesameBERT. Although our proposed architecture performed poorly in lower layers, it outperformed the other methods in higher layers. This finding indicated that in higher layers, using Squeeze and Excitation alongside Gaussian blurring helps self-attention models to capture global information in all layers.

### 4.3.2 HANS DATASET RESULTS

We trained both BERT and SesameBERT on the MNLI-m dataset to evaluate their classification accuracy. Similar to the results of another study (Devlin et al., 2019), BERT achieved $84.6\%$ accuracy, which is higher than that of SesameBERT, as shown in Table 1. In the HANS dataset, we explored the effects of two models on each type of heuristic. The results are presented in Figure 3; we first examined heuristics for which the label was Entailment. We can see that both models performed well; they assigned the correct labels almost $100\%$ of the time, as we had expected them to do after adopting the heuristics targeted by HANS.

Table 3: Comparison of specified layers among various approaches in the RTE dataset. We dissected our models into two methods. SE-BERT refers to BERT with Squeeze and Excitation; BLUR-BERT refers to BERT with Gaussian blurring.

| Layers | BERT | SE-BERT | BLUR-BERT | SesameBERT |
|---|---|---|---|---|
| Dev Set Accuracy | | | | |
| First Hidden Layer | 58.1 | 57.0 | **64.6** | 54.5 |
| Second Hidden Layer | 55.6 | 56.3 | **57.4** | 54.2 |
| Second-to-Last Hidden | 64.6 | 69.0 | 67.5 | **70.8** |
| Last Hidden | 65.3 | 67.9 | 68.6 | **70.4** |
| Sum Last Four Hidden | 65.0 | 69.3 | 68.2 | **69.7** |
| Sum All 12 Layers | 68.2 | 68.6 | 66.4 | **69.0** |
| Weighted Average Layers | 66.8 | 67.5 | 67.9 | **70.0** |

Next, we evaluated the heuristics labeled as Non-entailment. BERT performed poorly for all three cases, meaning that BERT assigned correct labels based on heuristics instead of applying the correct rules of inference. By contrast, our proposed method performed almost three times as well as BERT in the case of "Lexical overlap".

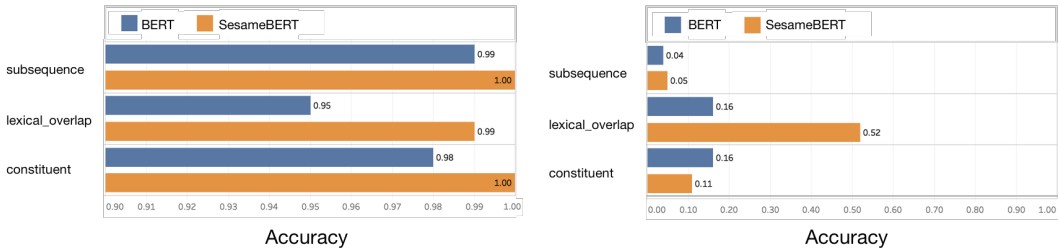

Figure 3: We compared BERT and SesameBERT for each case. **Left**: Results of heuristics-entailed cases. **Right**: Results of heuristics labeled as Nonentailment. In contrast to the results in **Left**:, BERT performed poorly in all three cases in **Right**; this indicated that the model had adopted shallow heuristics rather than learning the latent information that it intended to capture.

This paper argues that capturing local contexts for self-attention networks with Gaussian blurring can prevent models from easily adopting heuristics. Although our models performed poorly in cases of "Subsequence" and "Constituent", both of these heuristics may be hierarchical cases of the lexical overlap heuristic, meaning that the performance of this hierarchy would not necessarily match the performance of our models (McCoy et al., 2019).

## 5 CONCLUSION

This paper proposes a fine-tuning approach named SesameBERT based on the pretraining model BERT to improve the performance of self-attention networks. Specifically, we aimed to find high-quality attention output layers and then extract information from aspects in all layers through Squeeze and Excitation. Additionally, we adopted Gaussian blurring to help capture local contexts. Experiments using GLUE datasets revealed that SesameBERT outperformed the BERT baseline model. The results also revealed the weight distributions of different layers and the effects of applying different Gaussian-blurring approaches when training the model. Finally, we used the HANS dataset to determine whether our models were learning what we wanted them to learn rather than using shallow heuristics. We highlighted the use of lexical overlap heuristics as an advantage over the BERT model. SesameBERT could be further applied to prevent models from easily adopting shallow heuristics.

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

## A   DESCRIPTIONS OF GLUE DATASETS

Table 4: Descriptions of GLUE tasks. The second and third column denote the sizes of the corresponding corpora. All tasks are classification tasks, except for STS-B, which is a regression task.

| Corpus | #Train | #Test | Task | Metrics | Domain |
|--------|--------|-------|------|---------|--------|
| | | | *Single − Sentence Tasks* | | |
| CoLA | 8.5k | 1k | acceptability | Matthews correlation | misc |
| SST-2 | 67k | 1.8k | sentiment | Accuracy | movie reviews |
| | | | *Similarity/Paraphrase Tasks* | | |
| MRPC | 3.7k | 1.7k | paraphrase | Accuracy/F1 | news |
| STS-B | 7k | 1.4k | sentence similarity | Pearson/Spearman corr. | misc |
| QQP | 364k | 391k | paraphrase | Accuracy/F1 | social QA |
| | | | *Inference Tasks* | | |
| MNLI | 393k | 20k | NLI | Accuracy | misc |
| QNLI | 105k | 5.5k | QA/NLI | Accuracy | Wikipedia |
| RTE | 2.5k | 3k | NLI | Accuracy | Wikipedia |
| AX | - | 1.1k | NLI | Matthews correlation | news, paper, etc |

## B   DESCRIPTION OF HANS DATASET

Table 5: Three types of heuristics targeted by the HANS dataset. The examples show incorrect entailment predictions that would result from targeting these heuristics.

| Heuristic | Definition | Example |
|-----------|-----------|---------|
| Lexical overlap | Assume that a premise entails all hypotheses constructed from words in the premise | **The docter** was **paid** by **the actor**. $\xrightarrow[\text{WRONG}]{}$ The doctor paid the actor. |
| Subsequence | Assume that a premise entails all of its contiguous subsequences. | The doctor near **the actor danced**. $\xrightarrow[\text{WRONG}]{}$ The actor danced. |
| Constituent | Assume that a premise entails all complete subtrees in its parse tree. | If **the artist** slept, the actor ran. $\xrightarrow[\text{WRONG}]{}$ The artist slept. |

## C   LAYER WEIGHTS CALCULATED BY SQUEEZE AND EXCITATION

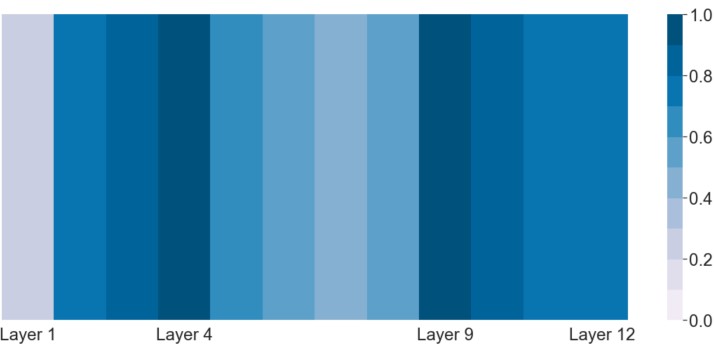

Figure 4: Evaluation of the weights calculated by Squeeze and Excitation for all layers, with the RTE dataset as an example.

