# OpenReview forum: "SesameBERT: Attention for Anywhere"
_ICLR.cc/2020/Conference — Reject_

### Official Review · AnonReviewer2 · 2019-10-23
**Official Blind Review #2**

**Rating:** 3

**Review:**

This paper proposes a novel BERT based neural architecture, SESAME-BERT, which consists of “Squeeze and Excitation” method and Gaussian blurring. “Squeeze and Excitation” method extracts features from BERT by calculating a weighted sum of layers in BERT to feed the feature vectors to a downstream classifier. To capture the local context of a word, they apply Gaussian blurring on output layers of the self-attention layer in BERT. The authors show their model’s performance on GLUE and HANS dataset.

Strengths
*This paper claims the importance of the local context of a word and shows an effect of their method on the various datasets: GLUE, and HANS.

Weaknesses
* It seems like the self-attention layer can learn the local context information. Finding important words and predicts contextual vector representation of a word is what self-attention does.
So, if using local-context information, which is information in important near words, is an important feature for some downstream tasks, then the self-attention layer can learn such important near words by training the key, query, and value weight parameters to connect the near important words.
It would be nice if the authors provide some evidence that self-attention can't learn such a local-context feature.

*In table 1, their experimental results show a slight improvement by using their method, but it's not significant.

* On HANS dataset, they show using local-context can prevent models from easily adopting heuristics. How Gaussian blurring can prevent that problem? More explanation about the relation between local-context and adopting heuristics is required.





**Experience Assessment:**

I have read many papers in this area.

**Review Assessment: Checking Correctness Of Derivations And Theory:**

I carefully checked the derivations and theory.

**Review Assessment: Checking Correctness Of Experiments:**

I carefully checked the experiments.

**Review Assessment: Thoroughness In Paper Reading:**

I read the paper at least twice and used my best judgement in assessing the paper.

---

> ### Author Response · Authors · 2019-11-11
> **Answer to Reviewer #2**
>
> We thank the reviewer for the detailed comments. In what follows, we address in detail the raised issues.
>
> 1. Thanks for your advice. We will do more comprehensive research in the future.
>
> 2、3. Like we've mentioned in official response.

---

### Official Review · AnonReviewer3 · 2019-10-23
**Official Blind Review #3**

**Rating:** 3

**Review:**

The paper proposes fine-tune methodologies for BERT-like models (namely, SeasameBERT).  This includes a method that considers all BERT layers and captures local information via Gaussian blurring. The methods were evaluated on several baseline datasets (e.g., GLUE, HANS)

Strengths:

* The paper is easy to follow.

*  Squeeze-and-extraction was used to incorporate all hidden layers instead of the common-practice of averaging last 4-layers. I find it both logical and useful.

* The suggested gaussian blurring method is able to capture local dependencies, which is missing in attention-based transformer layer.

*  SesameBERT improves performance on some GLUE metrics and on HANS dataset. Also ablation analysis suggests squeeze-and-extraction is a good technique to extract features from BERT model compared to other common practices.


Weaknesses:

* In my opinion, the paper novelty is not significant enough. Although useful, the suggested techniques are based on existing methods.

*  Incorporate spatial/context-information is usually done by concatenating a location-based embedding with the original word embedding. I’m curious if the blurring Gaussian will be as useful compared to such version.

* Since the suggested methods are generic, It can be more convincing to see results on recent models, and not only BERT. Currently, the results are not significantly better.

* The HANS DATASET RESULTS section seems rushed, will be good to elaborate more about HANS. also the first sentences of the section discusses GLUE results not HANS.

To conclude: The paper is easy to follow, suggests two nice methods for fine-tune BERT. But although useful, the suggested methods are not novel enough. The performance does not significantly improves, and the methods are applied only to BERT model.

**Experience Assessment:**

I have read many papers in this area.

**Review Assessment: Checking Correctness Of Derivations And Theory:**

I assessed the sensibility of the derivations and theory.

**Review Assessment: Checking Correctness Of Experiments:**

I carefully checked the experiments.

**Review Assessment: Thoroughness In Paper Reading:**

I read the paper at least twice and used my best judgement in assessing the paper.

---

> ### Author Response · Authors · 2019-11-11
> **Answer to Reviewer #3**
>
> We thank the reviewer for the detailed comments. In what follows, we address in detail the raised issues.
>
> 1、3. Like we've mentioned in official response.
>
> 2. Sorry, not pretty sure what you mean.
>
> 4. We first talked GLUE results on the section because we wanted to prove that the significant improvement on HANS dataset was based on a model with similar accuracy on GLUE tasks. We will highlight more on HANS dataset in the final version.

---

### Official Review · AnonReviewer1 · 2019-10-24
**Official Blind Review #1**

**Rating:** 3

**Review:**

Summary:
The paper proposes adding two mechanisms to the BERT architecture for NLU. The first is based on integrating information from all layers of the encoder via a method called Squeeze and Excitation. The second uses Gaussian blurring to encourage information sharing among neighboring words. The proposed method improves modestly on BERT on the GLUE suite of problems. It also substantially improves on BERT with respect to a class of examples that are designed to confound models that learn superficial heuristics based on word occurrence.

I learn toward rejecting this paper. The method shows some performance gains over BERT on some GLUE tasks, but these are fairly small for the most part, and BERT outperforms the proposed method by a similar amount on a similar number of tasks. The strongest result is the HANS "lexical_overlap" case, where the proposed method has a clear advantage. I have no experience with these kinds of NLU models, so I can't say with confidence whether the architectural additions proposed are well-motivated, but to me it feels like there is not a strong justification for adding these particular features to the BERT architecture, and the results do not clearly demonstrate their utility except in the "lexical_overlap" case.

Details / Questions:
* It seems to me that the GLUE results might be within the margin of error. Is it feasible to replicate training with different random seeds to see what the variance in the performance numbers might be? I suspect that a statistical analysis [1] might conclude that BERT and the proposed method are indistinguishable on the GLUE suite.

* Were the proposed architectural additions conceived with the HANS "counterexamples" in mind (i.e. is there a specific reason to think that these types of methods would avoid the "superficial" reasoning that these examples are supposed to reveal)? Were other methods of adding context considered?

* I suggest using the same x-axis scale on the two charts in Figure 3 to avoid confusion about the magnitudes of the differences.

References:
[1] Demšar, J. (2006). Statistical comparisons of classifiers over multiple data sets. Journal of Machine Learning Research, 7(Jan), 1-30.

**Experience Assessment:**

I do not know much about this area.

**Review Assessment: Checking Correctness Of Derivations And Theory:**

N/A

**Review Assessment: Checking Correctness Of Experiments:**

I assessed the sensibility of the experiments.

**Review Assessment: Thoroughness In Paper Reading:**

I read the paper at least twice and used my best judgement in assessing the paper.

---

> ### Author Response · Authors · 2019-11-11
> **Answer to Reviewer #1**
>
> We thank the reviewer for the detailed comments. In what follows, we address in detail the raised issues.
>
> 1. Most paper only revealed averaged results rather than the variance. We've released our code on the gitHub, you may run the variance in each GLUE tasks through our approaches. In our paper, we've run the results in 10 random seeds, and picking up the top 5 metrics then averaged them. Although the final GLUE score is not that significant, there are some obvious difference in some GLUE tasks.
>
> 2. Like we've mentioned in official response.
>
> 3. Because the results in left fig of Figure 3 are started from 0.9X, we rescale the x-axis to make it visible. We will edit the figure in final version, thanks!

---

### Author Response · Authors · 2019-11-11
**Official response to a common comment**

We thank the reviewer for the detailed comments. In what follows, we address in detail the raised issues. Here we explain some common questions.

1. In this paper, because the adjustment is on fine-tuning process related to BERT, the results on GLUE score are not that significant, although there are some significant improvement on some GLUE tasks. Here we propose to use a more comprehensive and innovative way in dealing with fine-tuning process. Also, we're applying this approaches not only on BERT but also other models, including XLNet, in the process. This comparison might be done in the future.

2. In the paper "Right for the Wrong Reasons: Diagnosing Syntactic Heuristics in Natural Language Inference", there are some detailed sentence examples in each evaluation metrics. In our paper, here we just show the superficial accuracy on HANS dataset, we will put the detailed results accuracy on each examples in the final version. In sum, we assume that blur method could prevent those sentences look similar but are in different meaning. We will do further research on how blur method influence the sentence accuracy variance in the future.

---

### Decision · Program_Chairs · 2019-12-19

**Decision:**

Reject

**Comment:**

This paper proposes a few architectural modifications to the BERT model for language understanding, which are meant to apply during fine-tuning for target tasks.

All three reviewers had concerns about the motivation for at least one of the proposed methods, and none of three reviewers found the primary experimental results convincing: The proposed methods yield a small improvement on average across target tasks, but one that is not consistent across tasks, and that may not be statistically significant.

The authors clarified some points, but did not substantially rebut any of the reviewers concerns. Even though the reviewers express relatively low confidence, their concerns sound serious and uncontested, so I don't think we can accept this paper as is.